# Social determinants of health, health behaviors, and general health among colorectal cancer survivors: A cross-sectional study

Youran Lee[iD][1]*, Susan G. Silva[iD][1,2], Rosa M. Gonzalez-Guarda[1]

1 Duke University School of Nursing, Durham, NC, United States of America, 2 Duke University School of Medicine, Durham, NC, United States of America

* yl799@duke.edu

## Abstract

### Purpose

Colorectal cancer is the second leading cause of cancer death globally, yet the extent to which social determinants of health (SDOH) and health behaviors contribute to disparities in general health status among colorectal cancer survivors (CRCS) is not well understood. This study aimed to identify SDOH associated with general health among CRCS and determine whether the number of current positive health behaviors mediated these associations.

### Methods

Cross-sectional data from 655 CRCS participating in the 2017, 2019, or 2021 Behavioral Risk Factor Surveillance System survey were analyzed. The outcome was poor/fair general health. SDOH included age, gender, race/ethnicity, education, marital status, employment, income, and three healthcare accessibility factors. Current positive health behaviors were fruit intake, current vegetable intake, regular exercise, non-smoker, and non-user of alcohol. Multivariable regression covarying for chronic conditions was used to examine the relationship between the SDOH, health behaviors, and general health.

### Results

The sample was primarily older adults (79.5%) and Non-Hispanic White (75.2%). Most had healthcare access (95.5%), with 39.5% having multiple chronic conditions and 32.6% reporting poor/fair health. Odds of poor/fair health were significantly greater among those unmarried (aOR = 1.90, 95% CI = 1.08, 3.33, p = .0263), unemployed (aOR = 1.92, 95% CI = 1.09, 3.38, p = .0243), and reporting multiple chronic conditions (aOR = 2.97, 95% CI = 1.46, 6.01, p = .0026). The number of current positive health behaviors did not mediate these associations.

**Data availability statement:** Data are held in a public repository in Center for Disease control and prevention (CDC) Behavioral Risk Factor Surveillance System, https://www.cdc.gov/brfss/annual_data/annual_2021.html.

**Funding:** This study was supported by the Holditch-Davis PhD Student Research Award from the Duke University School of Nursing for conducting the small research project.

**Competing interests:** The authors have declared that no competing interests exist.

## Conclusion

Family support and socioeconomic factors are important social contributors to general health disparities among CRCS. Establishing comprehensive social support systems is important to enhance the general health of CRCS.

## Introduction

Colorectal cancer (CRC) is the second leading cause of cancer death in the United States, with an estimated 154,270 new cases and 52,900 deaths anticipated in 2025 [1]. Despite more effective treatments and improved supportive care leading to longer survival for colorectal cancer survivors (CRCS) [2], many CRCS continue to experience poor health status and diminished quality of life (QoL) [3]. Social determinants of health (SDOH) (e.g., educational and job opportunities, income, housing, transportation, public safety, food security, social inclusion, nondiscrimination) drive significant health inequities that affect general health status and health behaviors among CRCS [4]. Racial and ethnic minoritized populations and individuals with lower socioeconomic status (SES) are disproportionately affected by colorectal cancer and face increased exposure to risk factors such as unhealthy diet, sedentary lifestyle, limited access to preventive measure (e.g., chemoprevention and screening), and inadequate follow-up of abnormal test results [5]. Black and Hispanic/Latino CRCS are more likely to receive substandard care compared to their White counterparts, contributing to worse outcomes [6]. These inequities are shaped by individual-, provider-, health system-, community-, and policy-level factors, highlighting the importance of understanding the complex interplay among races, ethnicity, SES, and healthcare utilization [7].

Health behaviors play a crucial role in improving outcomes, overall health, and QoL among cancer survivors [8,9]. Specific positive health behaviors that contribute to improved QoL and physical functioning include the consumption of vitamin supplements, vegetables, and fruits [10,11] and engagement in physical activity [12–14], whereas high alcohol consumption and smoking are linked to poorer health outcomes [11,15,16]. Although some studies have examined social and behavioral determinants of health among gastrointestinal cancer survivors [3] or focused on behavior risk factors among CRCS [14,17], to our best knowledge, no studies to date have explored the combined relationships of SDOH, positive health behaviors, and health status among CRCS.

### Conceptual framework

The conceptual framework for this study was guided by an integrated SDOH framework [4] that facilitates the translation of scholarly work on SDOH into evidence-based interventions aimed at addressing health inequities. SDOH are categorized by SDOH capital (e.g., structural factors) and SDOH processes (e.g., interactions between people and systems). SDOH capital comprises quantifiable resources and opportunities affecting health outcomes, for which allotment to individuals or groups

is primarily determined by social factors such as race, ethnicity, gender, and social support. SDOH process describes shaping the interactions among people, groups, institutions, or systems which impact health. Fig 1 illustrates this conceptual framework and demonstrates how it informed our study aims. In this study, SDOH variables including SDOH capital factors and healthcare accessibility factors are used for analysis. The new knowledge generated will enhance understanding of how SDOH function contextually (i.e., through exposure to health risks or protective factors) to influence health outcomes such as morbidity and mortality, and health inequalities.

## Aims and hypotheses

This study's overall goals were to (a) better understand (based on the conceptual framework) the relationship between SDOH (SDOH capital factors and healthcare accessibility factors) and general health status among adult CRCS, and (b) determine whether the number of current positive health behaviors (daily fruit intake, daily vegetable intake, regular exercise, non-smoker, non-user of alcohol) mediated these relationships among CRCS, covarying for number of chronic conditions. This study specifically aimed to (1) identify SDOH related to poor/fair health, and (2) determine whether the number of current positive health behaviors mediate SDOH-related poor/fair health relationships among CRCS. We hypothesize that (H1) specific SDOH factors (minoritized race and ethnicity, lower educational level, unmarried or unpartnered status, low-household income, unemployment status) will be significantly associated with poor/fair health among CRCS, and (H2) health behaviors will partially mediate the relationship between SDOH and poor/fair health among CRCS.

## Methods

### Study design

This cross-sectional, correlational study of 655 adult CRCS used existing data from the 2017, 2019, and 2021 Behavioral Risk Factor Surveillance System (BRFSS) [18]. The outcome of interest was general health, dichotomized as poor/fair health or not. The SDOH considered included seven SDOH capital factors (sociodemographic factors that affect measurable resources) and three healthcare accessibility factors. Number of current positive health behaviors was examined as a possible mediator of identified SDOH and general health relationships. Number of chronic conditions was a covariate in all

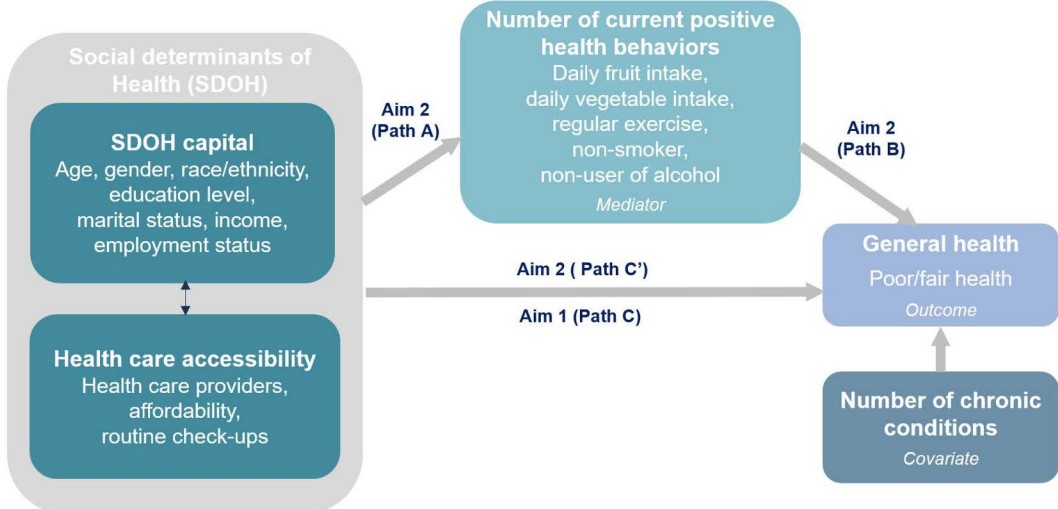

**Fig 1. Conceptual framework.**

analyses. This secondary analysis used existing publicly available BRFSS data and was reviewed and deemed exempt by the Institutional Review Board (IRB) at the university where the analysis was conducted.

## Data source

The BRFSS is a nationally representative, repeated cross-sectional study of U.S. adults, administered by state health departments and maintained by the Centers for Disease Control and Prevention (CDC) [18]. The BRFSS collects data on health-related behaviors, sociodemographic factors, leading preventable causes of death, and preventive health practices among non-institutionalized adult residents [3]. Data is collected via landlines or cellular telephones using a random digit dialing sampling technique [18]. The validity and reliability of BRFSS data have been well-documented [19]. Only data from the 2017, 2019, and 2021 BRFSS databases were included, as these were the only years in which survey questions about diet were asked.

## Sample

This analysis focused on individuals aged 18 and older who self-reported a history of colon or rectal cancer, identifying them as adult CRCS. Participants were excluded if they refused to respond to any survey questions or had missing or incomplete responses for any of the variables used in this analysis. The initial sample from 2017, 2019, and 2021 surveys included 1,306,977 participants aged 18 and older. After excluding 1,306,270 participants who had not been diagnosed with colon or rectal cancer, as well as those with missing sampling weights (n = 52), the final analytical sample consisted of 655 adults (Fig 2).

## Measures

***SDOH factors.*** This study included 10 SDOH variables that included measures of SDOH capital factors and healthcare accessibility factors. SDOH capital factors included the following seven factors: 1) age (categorized as 18–59 years or 60 years and older), 2) gender (male or female), 3) race/ethnicity (non-Hispanic White, Black/African American, Hispanic, or other races), 4) married/partnered (married, living with a partner, or unmarried including divorced, widowed, separated, and never married), 5) education level (high school graduate or less, bachelor's degree or some college, graduate degree), 6) employment status (unemployed, both partially and fully employed), and 7) annual household income (less than $25,000, $25,000 to less than $50,000, and $50,000 or more). Accessibility of health care variables were as follows: (8) having a healthcare provider, (9) needing but unable to afford to see a doctor, and (10) having had a routine checkup within the past year. The healthcare accessibility variables were coded as "yes" or "no."

***Chronic conditions (covariate).*** Clinical characteristics included 9 chronic conditions, each coded as 0 = *no/absent* or 1 = *yes/present*. Chronic conditions considered were the following: 1) heart attack/myocardial infarction, 2) angina, 3) stroke, 4) asthma, 5) chronic obstructive pulmonary disease (COPD), 6) depression, 7) kidney disease, 8) diabetes, and 9) arthritis. The total number of chronic conditions present were calculated (possible range: 0–9). In the analysis, the number of chronic conditions was included as a covariate and categorized into three levels: no chronic conditions, one chronic condition, two or more chronic conditions.

***Positive health behaviors***. The number of current positive health behaviors in which the participant was currently engaged was derived from five indicators (daily fruit intake, daily vegetable intake, regular exercise in the past month, non-smoker, non-user of alcohol. These variables were derived from a healthy lifestyle score related to CRC risk [20]. The five indicators were dichotomized and coded as follows: daily fruit intake (1 = *consuming fruit once or more times per day*, 0 = *not consuming fruit once or more times per day*); daily vegetable intake (1 = *consuming vegetables once or more times per day*, 0 = *not consuming fruit once or more times per day*), exercise regularly (1 = *participating in any physical activities or exercises such as running, calisthenics, golf, gardening, or walking for exercise during the past month*; 0 = *not*

*participating in any physical activities or exercises such as running, calisthenics, golf, gardening, or walking for exercise during the past month*), non-smoker (1 = *current non-smoker*, 0 = *current smoker*), non-drinker (1 = *not having any alcoholic drinks in the past 30 days*, 0 = *having at least one alcoholic drink in the past 30 days*). The total number of current positive health behaviors was calculated (possible range of 0–5) with higher scores indicating a greater current number of positive health behaviors.

*General health outcome.* The poor/fair health outcome (0 = *no*, 1 = *yes*) was derived from self-reported general health status using a 5-point Likert-scale. Absence of poor/fair health was defined as a general health status reported as *excellent*, *very good*, or *good*. Presence of poor/health was defined as general health reported as *fair* or *poor*.

*Design variables*. Design variables provided for 2017, 2019, and 2021 BRFSS databases were applied to generate weighted results that adjust for the complex survey design of each database. The design variables were applied to adjust for the under- and over-representation of subgroups in each database. Design variables included were the sampling weights, stratum variables, and cluster variables for the year in which the participant was surveyed.

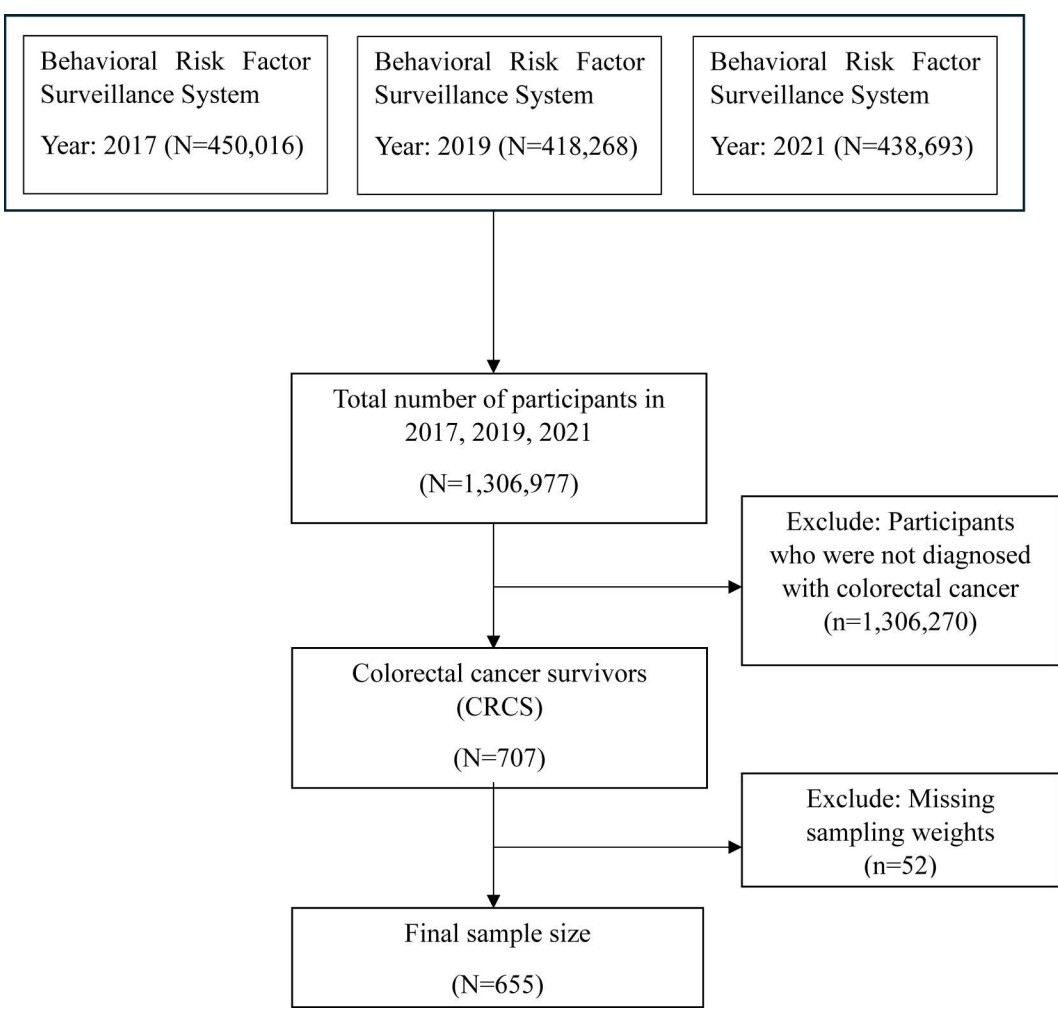

**Fig 2. Determination of the final analysis sample.**

## Statistical analysis

Unweighted and weighted descriptive statistics were used to detail the sample characteristics and key analytic variables. Non-directional statistical tests were performed with significance set at 0.05 per test. Effect sizes and their 95% confidence intervals (CIs) were performed to address clinical significance. Design variables were applied to all inferential statistical methods.

***SDOH and poor/fair health relationships (aim 1).*** First, separate bivariate logistic regression analyses were performed to examine the simple relationship between each of the 10 SDOH and number of chronic conditions with a poor/fair health outcome among the CRCS. Next, a full multivariable logistic regression model that included all 10 SDOH as key explanatory variables, number of chronic conditions as a covariate, and poor/fair health as the outcome of interest was conducted. Lastly, a backward elimination variable selection method was used to reduce the full multivariable model to a final parsimonious model that included only SDOH factors significant at the 0.05 level and number of chronic conditions as a covariate. Adjusted odds ratios (aORs) along with their 95% CIs were calculated to estimate effect size. Posteriori comparisons were conducted for explanatory variables with more than two levels when an overall significant effect of the explanatory variable on the outcome was detected ($p \leq .05$).

***Mediation analysis (aim 2).*** A traditional mediation approach [21,22] was used to determine whether the number of current positive health behaviors mediated the relationship between the set of SDOH in the final multivariable parsimonious model. To establish mediation, four path criteria were applied (see Fig 1). Path C examined the relationship between the SDOH identified as significantly associated with poor/fair health in the final parsimonious model, covarying for number of chronic conditions determined in the Aim 1 analysis ($p \leq .05$). The subsequent path analyses were performed to determine whether the number of current positive health behaviors mediated by the SDOH and poor/fair health relationship identified in the Aim 1 final parsimonious model (Path C). Path A tested for a significant relationship between the SDOH in the final multivariable parsimonious model and the number of current positive health behaviors (proposed mediator), using multivariable regression models covarying for number of chronic conditions. Effect sizes of the models were estimated using $R^2$ and their 95% CIs. Path B determined whether the number of current positive health behaviors was significantly related to poor/fair health, covarying for chronic conditions. Effect sizes were estimated using aORs and their 95% CIs. Path C' examined whether the effect sizes for the statistically significant relationships between SDOH and poor/fair health identified in the final multivariable parsimonious model Aim 1 (Path C) were diminished when the number of current positive health behaviors (proposed mediator) was included as a covariate in the model. Effect sizes were calculated using aORs to examine which SDOH effects were reduced when the proposed mediator was included as a covariate in the model.

## Statistical power

The sample size of 655 provided at least 80% power for the full multivariable logistic regression analyses with 11 explanatory variables to test for an statistically significant association between each SDOH and poor/fair health and the multiple regression models mediating effects of the number of positive health behaviors, assuming (1) two-tailed tests with significance set at 0.05 per test, and (2) small effects sizes (OR = 1.44 to 2.47) [23].

## Results

### Sample characteristics

Table 1 details the unweighted and weighted sample characteristics (N = 655) of the CRCS. Based on weighted results, most of the CRCS were aged over 60 years (79.5%), female (47.8%), and non-Hispanic White (75.2%). Approximately half of the participants (52.5%) held a higher education degree (bachelor's degree, some college or graduate degree), and 43.0% were unmarried (divorced, widowed, separated, or never married). Most were unemployed (66.4%) and had a household income less than $50,000 (55.6%). The majority had access to care, with 95.5% having a healthcare provider and 88.7% having

**Table 1. Sample characteristics.**

| Characteristic | N | Unweighted f | Unweighted % | Weighted % | 95% CI |
|---|---|---|---|---|---|
| **SDOH Capital Factors** | | | | | |
| Age category | 655 | | | | |
| 18 to 59 years | | 119 | 18.3% | 26.5% | 21.6-31.3 |
| 60 years or older | | 533 | 81.75% | 79.5% | 68.7-78.4 |
| Female Gender | 655 | 291 | 55.6% | 47.8% | 42.0-53.6 |
| Race/Ethnicity | 655 | | | | |
| Non-Hispanic White | | 544 | 83.1% | 75.2% | 69.8-80.6 |
| Non-Hispanic Black/AA | | 37 | 5.66% | 13.5% | 8.3-18.7 |
| Hispanic/Latinx | | 23 | 3.5% | 5.1% | 3.1-7.1 |
| Non-Hispanic Other Minorities | | 51 | 7.8% | 6.2% | 4.5-8.0 |
| Education | 655 | | | | |
| High school graduate or less | | 238 | 36.3% | 47.4% | 41.1-53.8 |
| Bachelor's degrees or some college | | 209 | 31.9% | 33.4% | 27.4-39.5 |
| Graduate degree | | 208 | 31.8% | 19.1% | 15.2-23.1 |
| Married/partnered | 654 | 342 | 52.3% | 57.0% | 51.0-63.2 |
| Employed | 653 | 172 | 36.3% | 33.6% | 28.0-39.2 |
| Annual household income | 559 | | | | |
| Less than $25,000 | | 155 | 27.7% | 24.9% | 20.5-29.2 |
| More than $25,000 to less than $50,000 | | 175 | 31.3% | 30.7% | 25.0-36.3 |
| $50,000 or greater | | 229 | 41.0% | 44.5% | 38.3-50.7 |
| **Healthcare Accessibility Factors** | | | | | |
| Having healthcare provider | 652 | 620 | 95.09% | 95.5% | 93.6-97.3 |
| Need to see the doctor but could not affordable | 654 | 38 | 5.81% | 6.2% | 3.4-9.0 |
| Having routine checkup less than 1 year | 648 | 566 | 87.35% | 88.7% | 85.1-92.3 |
| **Current Positive Health Behaviors** | | | | | |
| Daily fruit intake | 655 | 452 | 69.0% | 65.9% | 59.4-72.2 |
| Daily vegetable intake | 655 | 538 | 82.1% | 75.6% | 69.7-81.5 |
| Regular exercise | 655 | 430 | 65.7% | 65.0% | 58.9-71.0 |
| Non-smoker | 655 | 590 | 90.1% | 85.4% | 80.0-90.8 |
| Non-user of alcohol | 655 | 402 | 61.3% | 56.8% | 50.5-63.1 |
| **General Health** | | | | | |
| Poor/fair health | 655 | 221 | 33.7% | 32.6% | 26.8-38.3 |
| **Number of chronic conditions** | 655 | | | | |
| No chronic health conditions | | 185 | 28.2% | 33.0% | 27.3-38.7 |
| One chronic health condition | | 190 | 29.0% | 27.5% | 21.3-33.6 |
| 2 or more chronic health conditions | | 280 | 42.8% | 39.5% | 33.7-45.3 |

had a routine checkup in the past year. However, 6.2% reported that they were unable to afford to see a doctor when they needed to, and notably, 39.5% had two or more (multiple) chronic conditions. In terms of positive health behaviors, 65.9% consumed fruit daily, 75.6% consumed vegetables daily, and 65.0% reported having engaged in physical activities or exercises such as running, calisthenics, golf, gardening, or walking for exercise during the past month. Additionally, 85.4% were non-smokers, and 56.8% had abstained from alcohol in the past month. The mean of the number of positive health behaviors was 3.68 (SD = 1.00), with a median of 4.0 (25th, 75th percentile: 3, 4). Finally, 32.6% reported having poor or fair health.

 

**Aim 1. SDOH-poor/fair health relationship**

Table 2 shows the weighted bivariate results examining the association between number of chronic conditions and each SDOH factor with the poor/fair health (no/yes). There was a significant overall association of number of chronic conditions, marital status, and employment with poor/fair health (all p < .05). Specifically, the odds of having poor/fair health were significantly greater among those with two or more chronic conditions compared to those with no comorbidities (OR = 3.26, p = .0011), unmarried status compared to married status (OR = 2.01, p = .0120), and unemployed status compared to employed status (OR = 2.31, p = .0032).

Table 3 (Path C) presents the logistic regression results for the final multivariate parsimonious model for poor/fair health, covarying for number of chronic conditions. The odds of poor/fair health were significantly greater among those who were unmarried compared to those married/partnered (aOR = 1.90, p = .0263), those unemployed compared to those employed (aOR = 1.92, p = .0243), and those with two or more conditions compared to those with no chronic conditions (aOR = 2.97, p = .0026). This final model identified the SDOH significantly related to poor/fair health outcome after adjusting for chronic conditions that were used in the subsequent mediation analysis and represented Path C in that analysis.

**Aim 2. Mediation analyses**

Path A examined whether marital status and employment status were significantly related to the number of positive health behaviors (proposed mediator) using a multiple regression model, controlling for the total number of chronic conditions. The relationships between marital status and employment status with the number of positive health behaviors were not statistically significant after adjusting for chronic conditions (both: p > .05).

Path B examined whether the number of positive health behaviors was significantly related to poor/fair health, covarying for total number of chronic conditions, using a multivariable logistic regression model. No significant relationship between positive health behaviors and poor/fair health was detected (p = .6488); however, the number of chronic conditions was significantly related to poor/fair health (p = .0019). The odds of experiencing poor/fair health were three times greater among individuals with two or more chronic conditions compared to those with no chronic condition (aOR = 3.33, p = .0010); the odds of poor/fair health among those with one chronic compared to those with none did not significantly differ (p = .2855).

Path C' tested whether the effects of marital status and employment status on health were diminished when accounting for the effects of total positive health behaviors (proposed mediator). With total positive health behaviors as a covariate in the model, the relationship between marital status and employment with poor/fair health continued to be statistically significant (both p < .03), and the aORs were similar to those observed in the Path C (Aim 1) analysis for these variables. Consistent with the Path B results, total positive health behaviors was not related to poor/fair health in the Path C' model (p > .05).

Taken together, the results from the Path A, B, and C' analyses indicated that total number of current positive health behaviors (proposed mediator) did not partially or fully mediate the relationship between marital status and employment with poor/fair health.

## Discussion

This study was the first to comprehensively examine the combined relationships of SDOH, health behaviors, and general health status among CRCS specifically, thus addressing a critical gap in understanding of how SDOH and health behaviors influence the general health of CRCS. Our findings reveal that marital status and employment status are significant contributors to general health status among CRCS. Importantly, positive health behaviors did not mediate the relationship between these two SDOH and general health status. These results are crucial for nursing practice, as they emphasize the need to look beyond health behavior interventions and consider broader social context when addressing the health needs of CRCS.

 

**Table 2. Association between chronic conditions and SDOH with poor/fair health: Weighted bivariate results.**

| SDOH Characteristic | Poor/Fair Health = Yes n (%) | Odds Ratio (OR) | 95% OR CI | p-value |
|---|---|---|---|---|
| **Chronic conditions** | | | | .0024 |
| 2 or more chronic health conditions | 45.4% | 3.26 | 1.61-6.64 | .0011 |
| One chronic health condition | 28.8% | 1.57 | 0.67-3.73 | .2896 |
| No chronic health condition *(ref)* | 20.3% | | | |
| **Age category** | | | | |
| 18 to 59 years | 30.5% | 0.88 | 0.48-1.62 | .6770 |
| 60 years or older *(ref)* | 33.3% | | | |
| **Gender** | | | | |
| Female | 29.9% | 1.29 | 0.76-2.18 | .3480 |
| Male *(ref)* | 35.5% | | | |
| **Race** | | | | |
| Non-white | 33.5% | 1.06 | 0.57-1.97 | .8656 |
| White *(ref)* | 32.3% | | | |
| **Education** | | | | .4074 |
| High school graduate or less | 35.8% | 1.53 | 0.82-2.84 | |
| Bachelor's degrees or some colleges | 31.3% | 1.25 | 0.64-2.45 | |
| Graduate degree *(ref)* | 26.8% | | | |
| **Married/partnered** | | | | |
| Unmarried | 41.3% | 2.01 | 1.17-3.45 | .0120 |
| Married/partner *(ref)* | 25.9% | | | |
| **Employment status** | | | | |
| Unemployed | 38.3% | 2.31 | 1.38-4.03 | .0032 |
| Employed *(ref)* | 21.2% | | | |
| **Annual household income** | | | | .0873 |
| Less than $25,000 | 38.6% | 1.90 | 1.04-3.47 | |
| $25,000 to $49,99 | 37.0% | 1.77 | 0.92-3.42 | |
| $50,000 or greater *(ref)* | 24.9% | | | |
| **Has healthcare provider** | | | | |
| Yes | 31.7% | 1.12 | 0.55-2.29 | .7498 |
| No *(ref)* | 29.3% | | | |
| **Need to see the doctor but could not affordable** | | | | |
| Yes | 27.8% | 0.78 | 0.31-1.98 | .6048 |
| No *(ref)* | 32.9% | | | |
| **Routine checkup less than 1 year** | | | | |
| Yes | 31.3% | 0.83 | 0.39-1.75 | .6199 |
| No *(ref)* | 35.5% | | | |

Bivariate logistic regression results; 95% CI = 95% Confidence interval for odds ratio (OR)

In this study, the substantial impact of chronic conditions on health outcomes in both the general population and among CRCS underscored the necessity of accounting for these variables in future research. By carefully controlling for chronic conditions, researchers were able to more accurately isolate the effects of SDOH and health behaviors on general health status, particularly within the context of CRC; however, it is essential to acknowledge the complexities inherent in this

**Table 3. Final multivariable regression results.**

| Path | Outcome | Explanatory Variables | β (SE) | p-value |
|------|---------|----------------------|--------|---------|
| A | Positive health behaviors | Married/partner: Unmarried vs Married/partner | 0.20 (0.16) | .2203 |
| | | Employment: Unemployed vs Employed | −0.24 (0.18) | .1812 |
| | | Chronic conditions | 0.16 (0.09) | .0926 |
| | | 2 or more chronic health conditions | | |
| | | One chronic health conditions | | |
| | | No chronic health conditions | | |
| | | | **aOR (SE)** | **p-value** |
| B | Poor/fair health | Positive health behaviors | 0.95 (0.16) | .6488 |
| | | Chronic conditions | | .0019 |
| | | 2 or more chronic health conditions | 3.33 (0.36) | .0010 |
| | | One chronic health conditions | 1.61 (0.44) | .2855 |
| | | No chronic health conditions *(ref)* | | |
| C | Poor/fair health* | Married/partner: Unmarried vs Married/partner *(ref)* | 1.90 (0.29) | .0263 |
| | | Employment: Unemployed vs Employed *(ref)* | 1.92 (0.29) | .0243 |
| | | Chronic conditions | | .0044 |
| | | 2 or more chronic health conditions | 2.97 (0.44) | .0026 |
| | | One chronic health conditions | 1.43 (0.36) | .4163 |
| | | No chronic health conditions *(ref)* | | |
| C' | Poor/fair health | Married/partner: Unmarried vs Married/partner *(ref)* | 1.88 (0.28) | .0268 |
| | | Employment: Unemployed vs Employed *(ref)* | 1.94 (0.29) | .0256 |
| | | Chronic conditions | | .0035 |
| | | 2 or more chronic health conditions | 3.01 (0.36) | .0024 |
| | | One chronic health conditions | 1.44 (0.45) | .4165 |
| | | No chronic health conditions *(ref)* | | |
| | | Positive health behaviors | 0.95 (0.13) | .6999 |

Positive health behaviors = Number of current positive health behaviors (range: 0–5); β = Standardized regression coefficient; SE = standard error; aOR = adjusted odds ratio; aOR for positive health behaviors based on ascending order (0–5). **Path C addresses Aim 1,** while **Path A, B, and C' addresses Aim 2**, whether the number of current positive health behavior is a mediator of Path C relationship; Path A multiple regression model: overall $R^2$ = 0.0340.

population. Most CRCS are older adults, many of whom have multiple chronic conditions [24], and some with comorbidity have potentially curative treatment unnecessarily modified, compromising optimal care [25]. This reality demands a nuanced approach that goes beyond mere statistical control. Future studies of CRCS are encouraged to focus on developing interventions that account for this complexity, thereby allowing for more targeted and effective strategies to enhance overall health and well-being. More attention should be paid to CRCS who have chronic diseases, and effective interventions should be adopted based on this consideration.

In this study, being unmarried and experiencing unemployment as a CRCS were associated with developing poor/fair health status, a finding aligned with numerous studies highlighting the critical role of social support during the cancer journey. Marital status, often used as a proxy of social support, has been associated with enhanced mental and physical health that positively influences cancer recovery and overall survival rates [3,26–28]. Employment status is also a key component of social support, providing economic stability, access to health insurance, and a sense of purpose, all of which have been found to contribute to better health outcomes [29,30]. High levels of social support have consistently been associated with improved QoL, reduced symptom severity, and enhanced coping mechanisms during cancer treatment [31]. Moreover, research suggests that strong social support may mitigate cancer-related mortality and improve

physical and psychological symptoms, thus contributing to more effective symptom management [32,33]. These previous findings align with the SDOH conceptual framework, in which employment and marital status are identified as key elements of SDOH capital and social support represents SDOH process (4). These interwoven factors emphasize the complex, comprehensive nature of structural SDOH in shaping health outcomes, and highlight that social support strategies for CRCS are essential for achieving better health outcomes and advancing cancer equity.

In this study, the total number of positive health behaviors did not mediate the relationship between marital and employment status with general health status. This result may be partly attributed to the absence of standardized, comprehensive, and multidimensional measures in both SDOH and health behaviors. Our study included only five indicators of health behaviors, and these were dichotomized based on simple thresholds. The current criteria for assessing healthy behaviors (e.g., diet, smoking, alcohol consumption, physical activity) lack specificity and may lead to inaccuracies. For example, in this study, diet was considered healthy if an individual consumed fruits and vegetables at least once daily, without accounting for the variety, quantity, or nutritional quality of these foods. Similarly, smoking status was marked as healthy if the individual was currently a non-smoker, disregarding past smoking history that could still have health implications. Regarding alcohol consumption, the criterion was also overly simplistic: an individual was considered to have healthy behavior if they had abstained from drinking in the past 30 days, regardless of previous long-term alcohol use. Physical activity assessments did not account for the intensity, duration, or type of exercise, all of which are crucial for understanding its health benefits. These broad and insufficiently detailed criteria may have led to an incomplete or misleading evaluation of an individual's health behaviors, underscoring the need for more nuanced and comprehensive assessment tools.

Although health behaviors did not mediate the relationship between marital and employment status with general health status in this study, they remain important factors that influence overall quality of life and long-term survival among CRCS [34]. In particular, diet was identified as an important risk and protective factor influencing CRC risk and symptom management processes. In a previous study, a high intake of red meat and consumption of four or more alcoholic drinks per day, compared to a low intake of red meat and occasional or no drinking, was associated with an increased incidence of CRC; conversely, a higher intake of dietary fiber, calcium, and yogurt was inversely associated with CRC risk [35]. Additionally, diet has been found to be a significant factor associated with the severity and management of common CRC symptoms [36–41]. For example, in a study by Hou et al. [42], a diet rich in fruits, vegetables, whole grains, and lean proteins not only reduced risk for CRC but also helped to alleviate CRC symptoms and improve response to treatment. Because the colon and rectum are key organs in the digestive system, it is crucial to focus on diet behaviors in the CRCS population.

The authors acknowledge some important limitations to interpreting the findings of this study. First, the population under study in the BRFSS dataset is relatively small compared to the overall population, and the availability of diet-related variables is limited to the years 2017, 2019, and 2021. Second, there are significant missing values in the dataset for key variables such as insurance status, gender identity, and living location (metropolitan, suburban, rural), along with a lack of geospatial data such as zip code, which prohibited inclusion of these important SDOH factors in our analysis. Although the BRFSS began including SDOH-related variables such as employment stability, food security, housing security, utility security, and transportation access in 2022 (Centers for Disease Control and Prevention, 2023), it is necessary to incorporate other important SDOH variables related to structural racism and discrimination experiences. Third, most sample participants (75%) were White American with limited racial and ethnic diversity, thus restricting our ability to fully examine the influences of SDOH on health outcomes; therefore, marginalized populations were underrepresented, and a cautious interpretation of findings is recommended. Fourth, the data used in this study were collected during the COVID-19 pandemic (2021), which may have influenced participants' self-reported health status, health behaviors, and access to care. These contextual factors should be considered when interpreting the findings. Fifth, the analysis was limited to the nine chronic conditions available in the BRFSS dataset. While this allowed for consistency across participants, it excluded gastrointestinal conditions that may be particularly relevant in a colorectal cancer survivor population. This limitation may

affect the comprehensiveness of chronic condition burden assessment in this study. Lastly, the BRFSS is a self-report survey and susceptible to information bias and recall bias.

In this study, we examined the total number of current positive health behaviors as a proposed mediator; this approach allowed for a more straightforward analysis of the relationship between health behaviors and general health status. Future research may benefit from exploring distinct subgroups of CRCS with varying profiles of health behaviors through methods such as latent class/profile analysis, as these approaches could (a) provide deeper insight into how different patterns of health behaviors impact overall health, and (b) contribute to the identification and analysis of differences in health behavior subgroups. Additionally, as this was a cross-sectional study, observing longitudinal outcomes and tracking changes in health outcomes related to dietary behaviors over time was challenging. Future research should focus on longitudinal data analysis to better understand these dynamics. Lastly, it is important to note that as a significant portion of CRCS are older adults, many are likely retired or unemployed due to their age.

Healthcare providers, particularly nurses, play a pivotal role in delivering holistic care that addresses both the medical and social needs of CRCS. Interventions should extend beyond direct medical care to include support related to SDOH, such as marital stability and employment. Nurses can advocate for and facilitate essential social services, manage chronic conditions effectively, and coordinate comprehensive care strategies to improve overall health outcomes. Physicians can play a key role in identifying and addressing social barriers by screening for housing instability, food insecurity, and transportation challenges during routine visits. Moreover, they can foster interdisciplinary collaboration with nurses, social workers, and community organizations to tackle these systemic issues. Beyond traditional clinical roles, healthcare providers can engage in community partnerships to identify and address upstream SDOH and participate in advocacy efforts to influence policies that promote health equity.

## Conclusion

This study highlights the important role of SDOH in influencing health outcomes for CRCS, and the consequent need for nurses to go beyond behavioral interventions to address upstream SDOH; understanding and addressing these interconnected factors through a holistic approach to cancer care and continuous support in clinical and community settings can lead to improved general health outcomes for CRCS.

## Acknowledgments

The authors would like to thank Donnalee Frega, PhD, for their editorial assistance.

## Author contributions

**Conceptualization:** Youran Lee, Susan G. Silva, Rosa M. Gonzalez-Guarda.

**Formal analysis:** Youran Lee, Susan G. Silva.

**Methodology:** Youran Lee, Susan G. Silva.

**Supervision:** Rosa M. Gonzalez-Guarda.

**Writing – original draft:** Youran Lee.

**Writing – review & editing:** Youran Lee, Susan G. Silva, Rosa M. Gonzalez-Guarda.

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
