## [Decision Letter · Decision Letter 0]

13 Mar 2025

PONE-D-24-56636Social Determinants of Health, Health Behaviors, and General Health among Colorectal Cancer SurvivorsPLOS ONE

Dear Dr. Lee,

Thank you for submitting your manuscript to PLOS ONE. After careful consideration, we feel that it has merit but does not fully meet PLOS ONE’s publication criteria as it currently stands. Therefore, we invite you to submit a revised version of the manuscript that addresses the points raised during the review process.

We look forward to receiving your revised manuscript.

Kind regards,

Usama Waqar, M.B.B.S

Academic Editor

PLOS ONE

Reviewers' comments:

Reviewer's Responses to Questions

**Comments to the Author**

1. Is the manuscript technically sound, and do the data support the conclusions?

Reviewer #1: Yes

Reviewer #2: Yes

2. Has the statistical analysis been performed appropriately and rigorously? 

Reviewer #1: Yes

Reviewer #2: Yes

3. Have the authors made all data underlying the findings in their manuscript fully available?

Reviewer #1: Yes

Reviewer #2: No

4. Is the manuscript presented in an intelligible fashion and written in standard English?

Reviewer #1: Yes

Reviewer #2: Yes

5. Review Comments to the Author

Reviewer #1: 1) In the introduction, positive effects of consumption of fruits and vegetables have been repetitive, describe it once and avoid repetition

2) what is the rationale for choosing the 9 chronic conditions? patients with a chronic condition related to GIT, including crohn's, ulcerative colitis etc. may have poor health outcomes and should be accounted for.

3) History of any other type of cancer is a very important covariate that is not being evaluated

4) The categorization of comorbidities into 0,1,2+ is not a standard approach. using a standard comorbidity scale for example, Charlson index will be a good way of categorization

5) A lot of repetition of the same concept/finding through out the paper, avoid repetition to improve clarity for readers

6) It can be logically explained that patients with chronic conditions might have poor health outcome. it would be good to have a subgroup analysis of SDOH and positive health behaviors on patients with no comorbidities.

6) Future recommendation is focused on the role of nurses. Add more about how doctors can play a part during follow ups

Reviewer #2: The authors conduct a cross sectional study to identify SDOH associated with general heatlh among colorectal cancer survivors. The authors concluded that comprehensive social support is needed to enahnce general heatlh of colorectal cancer survivors.

I have the following comments:

1. The results of the study were obtained using a survey level data. This can lead to several biases due to inherest design of data collection and study design. This is a major limitation of the study an may limit the inferences drawn from the results.

2. Several SDOH components were found to be not associated with general health status. This should be addressed in the discussion section along with positive findings. What could be the reasons for such results?

Abstract:

No comment

Introduction:

1. The following statements needs a citation:

a. "SDOH, which encompasses factors such as educational and job opportunities, income, housing, transportation, public safety, food security, social inclusion, and nondiscrimination, drives health inequities and impacts general health status."

b. These inequities arise from differences in exposure to risk factors such as unhealthy diets and sedentary lifestyles, limited access to preventive measures like chemoprevention and screening, and inadequate follow-up of abnormal test results.

2. The authors did a good job of explaining the relationship between SDOH and cancer incidence, and subsequently SDOH, healthy behaviors, and outcomes among colorectal cancer survivors. However, a more succint introduction would improve readability for readers.

3. Please reconsider removing sub-headings from introduction section as per submission guidelines.

Methods:

1. The authors identified colorectal cancer survivors through BRFFS database as "patients who self reported a history of colon or rectal cancer". However, is it possible to ascertain whether these patients are survivors or under treatment or have a recurrence of the disease? This is an important aspect as it has significant role in how SDOH impacts helath outcomes.

2. Since the data related to positive health behaviors was self reported by patients using telephonic encounter. This data is variable and subject to bias depending on patient's recent beahviors. This is a subjective assessment and may significantly impact the results of current study.

3. The authors reported poor/fair health outcome (0=no, 1=yes) as a categorical variable derived from a 5-point Likert-scale. Such data recoding might cause issues with data analsis as data collection / organized may not be intended for such recategorization / analysis.

Results:

No comments

Discussion:

No comments

6. PLOS authors have the option to publish the peer review history of their article (what does this mean? ). If published, this will include your full peer review and any attached files.

**Do you want your identity to be public for this peer review?** For information about this choice, including consent withdrawal, please see our Privacy Policy .

Reviewer #1: No

Reviewer #2: No

---

## [Author Response · Author response to Decision Letter 1]

17 Apr 2025

I responded all the comments in attached response to reviewers table.

---

## [Decision Letter · Decision Letter 1]

15 Jun 2025

PONE-D-24-56636R1Social Determinants of Health, Health Behaviors, and General Health among Colorectal Cancer SurvivorsPLOS ONE

Dear Dr. Lee,

Thank you for submitting your manuscript to PLOS ONE. After careful consideration, we feel that it has merit but does not fully meet PLOS ONE’s publication criteria as it currently stands. Therefore, we invite you to submit a revised version of the manuscript that addresses the points raised during the review process.

Please see reviewer comments below.

We look forward to receiving your revised manuscript.

Kind regards,

Usama Waqar, M.B.B.S

Academic Editor

PLOS ONE

Journal Requirements:

Reviewers' comments:

Reviewer's Responses to Questions

**Comments to the Author**

1. If the authors have adequately addressed your comments raised in a previous round of review and you feel that this manuscript is now acceptable for publication, you may indicate that here to bypass the “Comments to the Author” section, enter your conflict of interest statement in the “Confidential to Editor” section, and submit your "Accept" recommendation.

Reviewer #3: All comments have been addressed

2. Is the manuscript technically sound, and do the data support the conclusions?

Reviewer #3: Yes

3. Has the statistical analysis been performed appropriately and rigorously? 

Reviewer #3: Yes

4. Have the authors made all data underlying the findings in their manuscript fully available?

Reviewer #3: Yes

5. Is the manuscript presented in an intelligible fashion and written in standard English?

Reviewer #3: Yes

6. Review Comments to the Author

Reviewer #3: The authors have written a well-structured and timely manuscript analyzing the associations between social determinants of health (SDOH), health behaviors, and self-reported general health in colorectal cancer (CRC) survivors using data from the BRFSS survey. The focus on self-rated health and its predictors is of growing importance, particularly as oncology continues to shift toward patient-centered outcomes. The use of a large, representative dataset and appropriate statistical methodology strengthens the credibility of the findings.

The authors have made a commendable effort to address prior reviewer comments, and the revised manuscript shows improvement in clarity, structure, and scope. However, the following issues still require attention:

• The use of 2021 BRFSS data, which coincides with the COVID-19 pandemic, may have significantly influenced self-reported health status, health behaviors, and access to care. The potential impact of the pandemic on the findings should be acknowledged and briefly discussed in the limitations section.

• A prior reviewer noted that the set of chronic conditions included in the analysis omits gastrointestinal (GIT)-related conditions, which may be particularly relevant in a CRC survivor population. While the authors clarified that they relied on the nine chronic conditions available in the BRFSS dataset, this limitation should be explicitly acknowledged in the Methods or Discussion section.

If these minor but important revisions are made, the manuscript has the potential to make a valuable contribution to the literature on cancer survivorship and public health policy.

7. PLOS authors have the option to publish the peer review history of their article (what does this mean? ). If published, this will include your full peer review and any attached files.

**Do you want your identity to be public for this peer review?** For information about this choice, including consent withdrawal, please see our Privacy Policy .

Reviewer #3: No

---

## [Author Response · Author response to Decision Letter 2]

23 Jun 2025

I attached the file 'response to reviewers'. Thanks.

---

## [Decision Letter · Decision Letter 2]

22 Jul 2025

Social Determinants of Health, Health Behaviors, and General Health among Colorectal Cancer Survivors: A cross-sectional study

PONE-D-24-56636R2

Dear Dr. Youran Lee,

We’re pleased to inform you that your manuscript has been judged scientifically suitable for publication and will be formally accepted for publication once it meets all outstanding technical requirements.

Kind regards,

Ola Sukkarieh, Ph.D., M.P.H, R.N.

Academic Editor

PLOS ONE

Additional Editor Comments (optional):

Reviewers' comments:

Reviewer's Responses to Questions

**Comments to the Author**

1. If the authors have adequately addressed your comments raised in a previous round of review and you feel that this manuscript is now acceptable for publication, you may indicate that here to bypass the “Comments to the Author” section, enter your conflict of interest statement in the “Confidential to Editor” section, and submit your "Accept" recommendation.

Reviewer #3: All comments have been addressed

2. Is the manuscript technically sound, and do the data support the conclusions?

Reviewer #3: Yes

3. Has the statistical analysis been performed appropriately and rigorously? 

Reviewer #3: Yes

4. Have the authors made all data underlying the findings in their manuscript fully available?

Reviewer #3: Yes

5. Is the manuscript presented in an intelligible fashion and written in standard English?

Reviewer #3: Yes

6. Review Comments to the Author

Reviewer #3: The authors have done an incredible effort to incorporate the comments from all reviewers while maintaining their own interpretation.

7. PLOS authors have the option to publish the peer review history of their article (what does this mean? ). If published, this will include your full peer review and any attached files.

**Do you want your identity to be public for this peer review?** For information about this choice, including consent withdrawal, please see our Privacy Policy .

Reviewer #3: No

---

## [Editor Report · Acceptance letter]

PONE-D-24-56636R2

PLOS ONE

Dear Dr. Lee,

I'm pleased to inform you that your manuscript has been deemed suitable for publication in PLOS ONE. Congratulations! Your manuscript is now being handed over to our production team.

Kind regards,

on behalf of

Dr. Ola Sukkarieh

Academic Editor

PLOS ONE